# Behavioral and Cortical Activation Changes in Children Following Auditory Training for Dichotic Deficits

**DOI:** 10.3390/brainsci14020183

**Published:** 2024-02-17

**Authors:** Deborah Moncrieff, Vanessa Schmithorst

**Affiliations:** 1School of Communication Sciences and Disorders, University of Memphis, Memphis, TN 38152, USA; 2Institute for Intelligent Systems, University of Memphis, Memphis, TN 38152, USA; 3Department of Radiology, Children’s Hospital of Pittsburgh, Pittsburgh, PA 15213, USA; vanessa.schmithorst@chp.edu

**Keywords:** dichotic, auditory processing, fMRI, children, perceptual learning, training

## Abstract

We report changes following auditory rehabilitation for interaural asymmetry (ARIA) training in behavioral test performance and cortical activation in children identified with dichotic listening deficits. In a one group pretest–posttest design, measures of dichotic listening, speech perception in noise, and frequency pattern identification were assessed before and 3 to 4.5 months after completing an auditory training protocol designed to improve binaural processing of verbal material. Functional MRI scans were also acquired before and after treatment while participants passively listened in silence or to diotic or dichotic digits. Significant improvements occurred after ARIA training for dichotic listening and speech-in-noise tests. Post-ARIA, fMRI activation increased during diotic tasks in anterior cingulate and medial prefrontal regions and during dichotic tasks, decreased in the left precentral gyrus, right-hemisphere pars triangularis, and right dorsolateral and ventral prefrontal cortices, regions known to be engaged in phonologic processing and working memory. The results suggest that children with dichotic deficits may benefit from the ARIA program because of reorganization of cortical capacity required for listening and a reduced need for higher-order, top-down processing skills when listening to dichotic presentations.

## 1. Introduction

Dichotic listening tests have always been an important part of standard audiologic assessment for auditory processing disorder (APD) [1,2] and have been the most used tests among U.S. audiologists for the past decade [3,4]. Since their first use to assess processing skills in split-brain patients [5,6], deficit patterns in DL scores have been a common finding [7], notably in children with listening, learning, and reading problems but no known neurologic impairment [8]. A large asymmetry between ear scores is characterized as amblyaudia and more symmetrically reduced ear scores are characterized as dichotic dysaudia [9,10]. Identification of amblyaudia or dichotic dysaudia depends upon matched score patterns from two DL tests, a standard that is more specific and stringent than current requirements for diagnosing APD [1,2].

Kimura’s structural theory of DL [11] attributed a large DL asymmetry in split-brain patients to absent neural transmission through the corpus callosum following verbal input to the ear that is ipsilateral to the listener’s language-dominant hemisphere. Development of the corpus callosum is prolonged into early adulthood [12] and there is evidence of structural differences among individuals with dyslexia [13], persistent stuttering [14,15], and speech sound disorder [16], as well as in the brains of children raised in poverty [17], but there is no direct evidence of corpus callosum deficits in children identified with amblyaudia or dichotic dysaudia. Reports of structural differences in the corpus callosum among individuals with diverse behavioral manifestations raise questions on whether these structural abnormalities are prenatal or derived from epigenetic or neural processing modifications through afferent and efferent processes during development.

An attentional model of DL [18,19,20] proposed that an interaural asymmetry like amblyaudia stems from stronger right ear performance in biased listeners whose contralateral language-dominant hemisphere is primed by the input of verbal information, drawing attention toward the right ear. When normal adult listeners selectively attend to dichotic stimuli in one ear only, neural activity increases in the contralateral posterior parietal cortex, superior temporal cortex and inferior frontal regions [21,22,23]. Similarly, directing attention to one ear typically increases performance on that side during a behavioral DL task [24]. During divided attention (free recall) DL tasks when the listener is instructed to repeat words presented to both ears, neural activation tends to be bilateral in normal adults [22], but when scores are averaged across directed right and directed left conditions, asymmetries tend to be comparable to those obtained during a free recall, divided attention condition. These results suggest that directed attention tasks may be useful to assess a listener’s ability to follow directions and attend to one or the other ear, but the free recall DL test produces a reliable index of asymmetry in most cases in less time [25,26].

DL tests also involve binaural integration skills related to spatial listening and localization. Binaural integration follows neural analysis and encoding of interaural differences in timing and intensity in the lateral and medial olivary complexes located in the auditory brainstem [27,28] through pathways that typically mature by age 5 to 6 years [29]. A combination of excitatory and inhibitory activity leads to enhancement of prominent signals within the listener’s auditory field by suppressing signals that arrive later or at lower intensities. Later maturation of these skills explains why children are unable to perform DL tests until at least age 5 or 6 years and why performance typically improves from then until late adolescence or early adulthood [30]. As children develop, DL scores increase in both ears, but asymmetry typically decreases following larger score increases in their non-dominant ears (usually the left ear).

A brainstem-to-cortex structural model of binaural integration proposes that greater symmetry in performance through development improves through (1) increased inhibition in dominant pathways or enhanced excitation in non-dominant pathways, leading to greater binaural symmetry [31] and (2) maturation and myelination of the auditory pathway, including portions of the corpus callosum, transmitting neural representations from the child’s non-dominant ear into the language-dominant hemisphere [32]. Brief, unilateral conductive hearing loss interferes with normal development in these brainstem-to-cortex pathways [33,34], suggesting that early factors limiting the availability and processing of sound could impede binaural integration and related processes within auditory pathways. The amblyaudia pattern was identified in 47% of children (n = 141) clinically tested for APD [8], in 12.4% of adolescents residing in a county detention center (n = 1158) and 8.8% among typically developing children aged 5 to 18 (n = 328) in the mid-Atlantic region [35]. The dichotic dysaudia pattern was identified in 31% of the children clinically tested for APD [8], 26.9% of adjudicated adolescents, and 12.5% of typically developing children. Emerging evidence suggests a similar prevalence for amblyaudia but a higher prevalence of the dichotic dysaudia pattern among typically developing school-age children in the mid-South [36]. These dramatically high rates for prevalence of dichotic deficits, only one of many auditory processing deficits that may interfere with communication and learning among school-age children, cast doubt on current estimates that only 2–7% of children suffer from any type of APD [37].

Auditory rehabilitation for interaural asymmetry (ARIA) is a short-term (4-week) auditory training regimen that was developed to remediate dichotic deficits by creating lists of dichotic material (words and digits) for presentation through sound field speakers to simulate naturalistic binaural listening conditions. Originally produced to improve listening in the non-dominant ear for individuals with the amblyaudia pattern, ARIA follows the principles of constraint-induced therapy (CIT) that have improved visual [38] and motor skills [39,40] following unilateral injuries. CIT involves intensive activity to force the use of the affected side (non-dominant ear) while constraining the unaffected side (dominant ear) through repetitive practice and individualized shaping to progressively improve performance. During ARIA sessions, the intensity of verbal input to the dominant ear is systematically varied to facilitate performance in the listener’s non-dominant ear throughout weekly dichotic listening sessions.

In the first clinical trials of ARIA, children with asymmetric dichotic deficits demonstrated improvements in DL performance not observed in a control group of children [41]. In a multisite study designed to establish the efficacy of ARIA, children with both deficit patterns improved, but those with the amblyaudia pattern demonstrated the largest improvements in DL with large effect sizes (non-dominant ear = 0.79 and 0.73 and advantages = 0.66 and 0.68 for words and digits, respectively) [42]. We hypothesize that the larger gains in the non-dominant ear from ARIA are likely due to adaptive, individualized adjustments to interaural intensity that induce synaptic alterations in the central auditory system. The model for ARIA proposes that by constraining the dominant ear, we increase inhibition in brainstem auditory pathways from that side so that when the listener is instructed to repeat the words, excitation through the non-dominant auditory pathways strengthens with practice. Our goal is to facilitate greater binaural symmetry in abnormally asymmetric systems [31]. We further hypothesize that changing the balance between excitation and inhibition in the auditory brainstem may induce neuroplastic mechanisms of Hebbian perceptual learning [43] that promote more efficient coding of verbal signals through ascending auditory pathways into the cortex. The ARIA protocol includes rest, known to improve memory consolidation of newly learned tasks [44], in the middle of the ARIA training session and between the weekly sessions. Our goal is to enhance access to bottom-up binaural verbal signals in listeners suffering from dichotic deficits to aid their ability to process speech-based signals under challenging everyday listening circumstances. Processing of speech involves activation in bilateral temporal regions, including the posterior superior and middle temporal gyri and the superior temporal sulcus [45,46,47,48]. When most listeners engage in a divided attention, free recall DL task like the kind used to identify these deficits, there is a small leftward bias in these cortical areas [49,50,51,52] and in frontal regions involved in executive control [53,54,55], presumably from a structural advantage between the right ear and language-dominant left hemisphere. Some listeners, however, may produce the atypical pattern with better scores in their left ear and are likely to have language dominance in their right hemisphere.

This study’s main purpose was to compare behavioral DL scores and functional MRI activation levels before and after participation in ARIA. We hypothesized that children with dichotic deficits, especially those with amblyaudia, may demonstrate a larger leftward bias in their cortical activation patterns from stronger input from their dominant right ears. As a pilot study to explore the neurophysiologic effects from ARIA, we hypothesized that the therapy would ease the challenges of processing dichotic stimuli in these children, reflected in better DL scores in their non-dominant ears and reduced levels of activation during dichotic listening conditions in the scanner.

## 2. Materials and Methods

### 2.1. Participants

Seventeen children were recruited to participate in the study following clinical identification of dichotic deficits. Parents provided signed consent forms as approved by the Institutional Review Board of the Human Research Protection Office of the University of Pittsburgh and by the University of Pittsburgh Medical Center Children’s Hospital. Two participants were removed from the study following the first fMRI scan, in one case due to excessive artifact from motion during acquisition, and in the other case following identification of an abnormality, as required by the IRB-approved protocol. Six participants were unable to receive their final fMRI scan because the hospital suspended all research imaging procedures for several months, unrelated to our study protocol. The final study included 9 children—4 males and 5 females—ranging in age from 7 to 13 years (mean = 8.8 years).

### 2.2. Baseline Behavioral Tests

A parent completed an auditory processing checklist comprised of 20 questions, each related to auditory difficulties associated with auditory decoding, integration, and prosody categories as characterized by the Bellis–Ferre model of auditory processing disorder [56,57]. The parent assigned a score of 0, 1, 2, or 3 for “never”, “now and then”, “often”, and “always” in response to questions regarding his or her child’s auditory behaviors, such as “requested rephrasing of verbal messages”, “needed reminders about ‘how to do’ things”, and “responded inappropriately to verbal messages”. If a question was not applicable, the parent could leave it blank. The parent’s responses were totaled within each category and divided by the maximum score possible for the number of questions answered to derive a percentage for each category.

All testing was performed through inserted earphones attached to a standard clinical audiometer. Pure tone hearing thresholds were measured in each ear at 500, 1000, 2000 and 4000 Hz, and the randomized dichotic digits test (RDDT) [58] and the dichotic words test (DWT) [59] were presented at 50 dB HL to each ear. Some children were also tested in the same manner with the “Competing Words” subtest from the SCAN [60]. The number of correctly identified words in each ear was tallied and converted to percentage correct. The ear with the higher score was identified as the “dominant ear” and the other as the “non-dominant ear”. If the scores were equal in the two ears for one test, dominance was assigned from the results of the other test. Interaural asymmetry was measured as the difference in performance between the two ears for each test. Results from the 2-pair condition of the RDDT and one list of 25 pairs from the DWT were compared to low cutoff scores representing the 5th, 10th, and 25th percentiles, respectively, from a 99.7% confidence interval for the participant’s age group. Each individual score was color-coded for severity, with the 5th percentile in red for severe, 10th percentile in yellow for moderate, and 25th percentile in green for minimal. Results from the Competing Words subtest for ear advantage during the right ear-first and left ear-first conditions were compared to prevalence data, and if a participant’s score was atypical, it was assigned a severity rating based on its prevalence, i.e., 2% was severe, 5% was moderate, and 10% was minimal. The pattern of the test results was identified as consistent with amblyaudia, dichotic dysaudia, or mixed for amblyaudia on one test and dichotic dysaudia on the other test. A final overall severity was identified in percentile rank for each participant [61].

Participants were evaluated for speech-in-noise performance with the Words-in-Noise test [62] and for temporal and tonal processing skills with the Frequency Pattern Test (FPT) in the verbal labeling condition [63]. Stimuli for both tests were presented binaurally through inserted earphones attached to a clinical audiometer. Individual scores were compared to available normative data for both tests.

### 2.3. fMRI Scanning

Before ARIA training began, participants received a pretreatment fMRI scan. All scans were acquired on a Siemens 3T Skyra system at the Department of Radiology in Children’s Hospital of the University of Pittsburgh Medical Center (UPMC). Standard 2-D EPI sequences were used with TR/TE = 2000/35 ms, 4 mm × 4 mm in-plane resolution and 4 mm slice thickness, with sufficient slice coverage for the whole brain. The HUSH (hemodynamics unrelated to scanner hardware used for silent-gradient acquisition) technique was used for presentation of auditory stimuli, consisting of a 5 s silent period for stimulus presentation, followed by 6 s of data acquisition (3 TRs). The fMRI stimuli were presented using Presentation software, version 20.0. The auditory stimuli consisted of one, two, or three numbers presented diotically (same word to both ears) or dichotically (different words presented to each ear), randomized together with periods of silence. Participants were instructed to listen to the numbers and silently repeat them. The order of stimulus presentation was randomized at runtime.

### 2.4. ARIA Training

Shortly after being scanned (mean interval = 23.5 days), each participant attended four weekly one-hour sessions of ARIA. Training sessions were conducted at the Auditory Neurophysiology Laboratory at the School of Health and Rehabilitation Sciences of the University of Pittsburgh. Each session included two 20 min training periods separated by a 20 min rest break. During the training, dichotic material (single-syllable words, digits, spondaic words, and fairy tale segments) was presented through sound-field speakers located at ±90° relative to the participant’s head. Presentations to the non-dominant ear were set at 50 dB HL while the material presented to the dominant ear was adjusted up and down in intensity in accordance with the relative performance between the two ears after each word list was presented. The intensity of presentations to the participant’s dominant ear was raised whenever performance in the non-dominant ear was better by more than 10% than performance in the dominant ear and lowered when non-dominant ear performance 10% or more lower than performance in the dominant ear.

### 2.5. Post-ARIA Behavioral Measures

Participants returned for a repeat of baseline behavioral measures with the RDDT, DWT, WIN and FPT 3 to 4.5 months following the end of ARIA training (mean interval = 3.5 months).

### 2.6. Behavioral Data Analysis

Scores from the auditory processing checklist were compared by multivariate analysis of variance (ANOVA) to an age- and gender-matched group of children whose DL results from a clinical evaluation for auditory processing disorder were normal. Individual ear scores for each DL test at baseline were compared by paired sample *t*-tests with Cohen’s d analysis of effect size. Average group scores for both DL tests and other auditory processing test measures at baseline were compared to the same scores after ARIA treatment by multivariate analysis of variance (ANOVA) with the coefficient of determination, r^2^, as a measure of effect size. All analyses were done in SPSS, v. 28 with the significance level for all comparisons at *p* < 0.05.

### 2.7. fMRI Analysis

Motion correction was performed using an affine transformation and a pyramid iterative routine, separately for the 1st, 2nd, and 3rd scans after the stimulus, as the baseline signal is different due to differences in longitudinal relaxation. Using SPM8, the EPI images were aligned into MNI space. A study-specific template was performed by averaging across all participants. The fMRI data were then transformed into standardized space using the study-specific template.

Separate analyses were performed for the contrasts of diotic vs. silence and dichotic vs. diotic. For each contrast, the 1st, 2nd, and 3rd scans after the stimulus were analyzed separately using a general linear model, with baseline drift (linear and quadratic) as nuisance regressors and active vs. baseline contrast as the regressor of interest. The magnitude (active vs. baseline % difference) and its standard error was stored for across-subject analyses.

An omnibus T-score was found for each contrast for each participant by taking the precision-weighted averages for the contrasts from the 1st, 2nd, and 3rd scans after the stimulus. A one-sample voxel wise *T*-test was then used to find regions of significant group activation for each contrast both before and after the training. Results were deemed significant at family-wise error (FWE)-corrected *p* < 0.05, using the Monte Carlo method after converting T-scores to Z-scores. Also, the difference in scans before and after training was computed (along with standard errors) and omnibus T-scores found in a similar manner. A one-sample *T*-test was again used to find regions of significant group differences in activation after vs. before training.

## 3. Results

### 3.1. Behavioral Results Pre-ARIA

Parents initially brought participants to the laboratory for an auditory processing assessment because of concerns regarding school performance. When asked to rate their child’s difficulties on the auditory processing checklist, parents reported the greatest difficulties with skills in the auditory decoding and integration categories, with average scores of 54% and 57%, respectively, and fewer difficulties with prosodic skills, scored at 39%. Compared to results produced by parents of an age- and gender-matched group of nine children whose DL scores were normal, parents of participants in this study reported higher scores, reflecting more listening difficulties in the auditory decoding category (F(1,17) = 10.1, *p* = 0.006, r^2^ = 0.39), integration category (F(1,17) = 13.0, *p* = 0.002, r^2^ = 0.45), and prosodic category (F (1,17) = 4.4, *p* = 0.053, r^2^ = 0.22).

All participants produced normal pure-tone thresholds and there were no interaural threshold differences greater than 5 dB HL.

All participants produced a right-ear advantage across DL measures. Six were identified with amblyaudia (two moderate, four severe), two were identified with dichotic dysaudia (one moderate, one severe), and one was identified as mixed, with scores consistent with amblyaudia (severe) on one test and dichotic dysaudia (severe) on the other test. As shown in Figure 1, average scores in dominant right ears and non-dominant left ears were significantly different during all three tests: RDDT, t(8) = −8.19, *p* < 0.001, Cohen’s d = 1.51; DWT, t(8) = −4.699, *p* = 0.002, Cohen’s d = 1.83; CW, t(8) = −6.736, *p* = 0.001, Cohen’s d = 1.84.

Two participants produced high signal-to-babble ratios (above the 90th percentile cutoff for age) for identifying 50% of the words presented in the WIN, as shown in bold in Table 1 Two others produced borderline scores that are shown in italics on Table 1. The WIN scores were negatively correlated with the non-dominant ear scores from the RDDT Pearson correlation (−0.710, *p* = 0.032) and positively correlated with the interaural asymmetry from the RDDT Pearson correlation (0.885, *p* = 0.002).

Percentage correct scores from the FPT that were below normal for all but two of the participants are also shown in bold on Table 1.

### 3.2. Behavioral Results Post-ARIA

Participants were reassessed with the RDDT, DWT, WIN and FPT following ARIA training. All pre-ARIA and post-ARIA behavioral results are shown in Table 1. Groupwise results demonstrated significant improvement in non-dominant ear performance during the RDDT (F(1,17) = 8.40, *p* = 0.010, r^2^ = 0.30) and DWT (F(1,17) = 6.99, *p* = 0.018, r^2^ = 0.30), as shown by the difference between the gray and black bars on the left side of Figure 2, but there was no groupwise change in dominant ear performance following ARIA treatment. As intended, there was also a significant reduction in interaural asymmetry across the group for the RDDT (F(1,17) = 35.71, *p* < 0.001, r^2^ = 0.69) and DWT (F(1,17) = 6.15, *p* = 0.025, r^2^ = 0.28), as shown in the right side of Figure 2. In addition to the significant change in DL non-dominant ear performance that was specifically targeted by the treatment protocol, the group average level for 50% performance on the WIN was 2.9 dB S/B lower following ARIA treatment (F(1,17) = 12.2, *p* = 0.003, r^2^ = 0.43). There was no groupwise change in performance on the FPT in the verbal labelling condition following ARIA treatment (F(1,17) = 0.22, *p* = 0.643).

Six of the participants produced DL scores that were no longer consistent with a deficit in the post-ARIA assessment. One participant (2114) who had been identified with moderate amblyaudia produced scores consistent with severe dichotic dysaudia after ARIA and also produced poorer scores on all other measures at the follow-up appointment. Because the parent reported that the participant was not being fully cooperative, a follow-up appointment was recommended, but the participant did not return.

Another participant (2183) with severe dichotic dysaudia prior to ARIA produced a severe mixed pattern, with severe dichotic dysaudia on the digits test and severe amblyaudia on the words test after ARIA, with little evidence of general benefit for dichotic listening. A participant who initially produced a pattern consistent with severe amblyaudia (2128) showed improvement in the non-dominant ear for both tests, but demonstrated a moderate deficit in the dominant ear on the digits test after ARIA.

### 3.3. fMRI Results Pre- and Post-ARIA

During diotic listening in the scanner, participants were listening to binaural presentations of one single number at a time. Pre-ARIA treatment, activation during diotic listening was compared to activation while lying in the scanner with no auditory stimulus presented to demonstrate the change in activation when listening binaurally to speech signals. As expected, group activation was seen in the superior temporal gyrus bilaterally, as shown in Figure 3. Group deactivation was seen in the precuneus, as shown on the left side of Figure 4, as well as other visual association areas, likely related both to typical posterior default node network (DMN) deactivation after performance of a task as well as possible visual processing of the scanner surroundings. When the activation seen during the same contrast (diotic versus silence) before ARIA was subtracted from the activation post-ARIA, there was greater activation post-ARIA in the anterior cingulate/medial prefrontal region, as shown on the right side of Figure 4.

During dichotic presentations in the scanner, participants listened to simultaneous presentations of two different numbers in each ear. A comparison between dichotic and diotic presentations before ARIA treatment (dichotic minus diotic) demonstrated more activation in the dichotic condition in the left posterior superior temporal gyrus, inferior frontal gyrus (Broca’s area; BA 44/45), and superior anterior cingulate, as shown on the left side of Figure 5. A small increase in activation was also seen in the right superior temporal gyrus. When pre-ARIA activity for the dichotic-diotic comparison was subtracted from post-ARIA activity, reduced activation, as shown on the right side of Figure 5, occurred in the primary auditory cortex bilaterally extending into the precentral gyrus, the RH homologue of BA 45 (pars triangularis), RH inferior temporal gyrus, and the RH dorsolateral and ventrolateral prefrontal cortex.

## 4. Discussion

Auditory training is not a new idea. In 1954, Myklebust’s “training” to remediate “peripheral deafness, aphasia, psychic deafness, or mental deficiency” in children was a recommendation for parents and teachers to use adaptive strategies to manage these poorly understood disorders [64]. Throughout the second half of the 20th century, adult patients with brainstem lesions and non-lesioned children with similar auditory processing test results were also advised to adapt to their listening difficulties with preferential seating and environmental modifications [65,66]. After the “decade of the brain” in the 1990s, recommendations emerged for auditory training based on the principles of neuroscience with bottom-up (tones and speech) and top-down (language) stimuli to improve listening comprehension, language processing and educational achievement [67]. It was argued that training should be intensive and adaptive, with feedback and reinforcement, targeted toward the specific deficits commonly identified through standard APD batteries, i.e., temporal processing, discrimination, closure, and binaural integration (dichotic listening). Despite the call for greater specificity in diagnosis and treatment, clinicians generally opted to continue using a battery approach to make the nonspecific diagnosis of APD and recommend a boilerplate list of general accommodations, a practice that has been widely criticized [68,69].

Given that many children with learning difficulties fail dichotic tests, a method to identify a dichotic deficit and provide specific bottom-up intervention to facilitate dichotic skills is a reasonable option for this population. Originally created in 2000 to provide dichotic training to children with abnormal DL interaural asymmetries, ARIA has led to improvements in symmetric and asymmetric dichotic deficits that did not regress over time and to higher scores for listening comprehension, word recognition and oral reading [41]. Since 2010, the ARIA protocol has been standardized for delivery by a trained clinician over four one-hour appointments to accommodate the needs of children throughout the calendar year [42].

There is growing evidence of other dichotic training benefits for children with auditory processing difficulties [70,71,72], including children with autism [73,74], but ARIA produces benefits with the lowest time in training. Rapid improvements in learning have often been attributed to “training to the task” and discounted as poor evidence of substantive changes in perceptual processing, but when evaluated in a frequency discrimination task, rapid gains in performance were noted to stem from perceptual rather than procedural learning [75]. Effective perceptual learning exhibits an early, rapid adaptation to the stimuli and procedures involved in the task and a later, slower, more gradual growth that promotes long-term retention and consolidation [76,77]. Evidence of long-term synaptic and intrinsic plasticity in auditory brainstem nuclei in animal studies, together with evidence of brainstem modifications following auditory training protocols, supports an emerging brainstem model for cellular changes that underlie learning behavior [78]. The benefits observed following short-term, intensive dichotic training in this study are consistent with a perceptual learning model that facilitates rapid synaptic adaptation to exaggerated interaural intensities at initiation of training, followed by parallel engagement of cortical working memory and attention networks to sustain and increase improvements over the full training interval. This view of ARIA is consistent with the reverse hierarchy theory (RHT) model of perceptual learning that when challenging speech is encountered, cortical perception fails and listeners perform a “backward search” to low-level acoustic features that will then be fine-tuned to enhance perception [79]. Fine-tuning purportedly occurs by reassigning weights to relevant and irrelevant properties in incoming stimulus features to enhance cues that will improve perception, a process long understood to occur when listeners adjust to different talkers [80] that may also be part of normal speech encoding.

Efforts to disambiguate subcortical and cortical neurophysiologic alterations during perceptual learning remain elusive, but the evidence in these children of neuroplastic changes in primary sensory areas followed by changes in anterior temporal and prefrontal areas over several training sessions supports the view that the right temporal cortex is important for parsing acoustic cues for perceiving concurrent events during challenging listening tasks [81]. In this case, immediate, short-term benefits would stem from sensory-related changes in primary cortical areas and sustained, long-term benefits redounding from practice that depends on a frontal system of working memory and attention mediated by the anterior cingulate, prefrontal cortex and associative striatum [82]. The improvements in non-dominant ear performance during dichotic listening tasks and reductions in functional activation observed in the right-side dorsolateral prefrontal cortex of the children in this study are consistent with this model of perceptual learning and with other emerging evidence of neuroplastic changes in the same regions following speech perception training.

Before training, these children demonstrated normal bilateral hemispheric activity when listening binaurally to the same word, with slightly higher levels of activity in the left superior temporal gyrus [83,84]. They also demonstrated deactivation of the precuneus, a part of the default-mode network (DMN) that is typically deactivated during cognitive task performance to reduce interference from self-referential thought [85], suggesting that they were able to sustain attention to perform a standard binaural listening task [86]. After ARIA, activation levels were lower in the anterior cingulate and medial prefrontal regions, areas that fulfill multimodal functions needed for the temporal processing of sensory events [87] and are sensitive to inputs from the medial and lateral superior olives of the brainstem [88]. This post-ARIA reduction in activity noted in the DMN’s medial prefrontal area during binaural listening could indicate that ARIA training enhanced the processing of bilateral sensory events ascending from the brainstem that contribute to fluctuations in attentional control during auditory tasks.

Dichotic listening normally activates a temporofrontal network that includes Heschl’s gyrus, the planum polare and planum temporale, the anterior and posterior superior temporal sulcus and the inferior frontal gyrus [89]. Greater activity is typical for dichotic compared to diotic stimuli in cingulate cortex [90] and during divided rather than selective attention tasks in the pre-supplementary motor area [91]. The supposition is that listening to dichotic stimuli places greater demands on attention, engaging more cognitive control to process information in the listener’s non-dominant ear (usually the left ear) [92,93,94]. Before ARIA, participants demonstrated similarly greater activation in the left posterior superior temporal gyrus, inferior frontal gyrus (Broca’s area; BA 44/45), and superior anterior cingulate when listening to dichotic compared to diotic numbers. The higher activity levels observed among these children, most of whom produced large interaural asymmetries during DL tasks, were observed in adults when selectively attending to dichotic CV stimuli in cognitively demanding conditions created by larger interaural intensities [95]. A comparison of children with large interaural asymmetries to a control group is needed to determine whether the activity levels among children with these dichotic deficits are higher than would be expected just by listening to presentations of dichotic numbers randomly interspersed with diotic numbers and silence.

After ARIA, significant reductions in activity during dichotic tasks were evident in bilateral auditory cortices and in the precentral gyrus, the site of the primary motor cortex. Reductions in the anterior cingulate could have been due to less need after ARIA for recruitment of attentional resources involved in motor rehearsal for speech production, often seen when individuals ignore distractors to remain focused on challenging tasks [96]. Since higher activity in the left precentral gyrus is associated with longer reaction times and may be an index of working memory demands during challenging tasks [97], it is plausible that these children required fewer resources for working memory and/or engaged in less subvocal rehearsal during dichotic listening tasks after ARIA treatment.

Deactivation also occurred after treatment in the right-hemisphere pars triangularis, a region of the inferior frontal gyrus known to be involved in semantic processing that may be necessary only when cognitive control is required to resolve linguistic conflicts. Interestingly, suppression of the right pars triangularis with transcranial magnetic stimulation resulted in improved naming in adult aphasic patients [98], suggesting an association between reduced activity in this region in the inferior frontal gyrus and improved access to linguistic representations. The pars triangularis contributes primarily to phonological processing [99] and is thought to be a human equivalent of a bilateral mirror neuron system that activates during production and perception of similar action [100]. It has also been implicated in semantic processing of language, as evidenced by larger N400 event-related potentials in response to anomalous, mismatched stimuli in sentence-level tasks [101]. When children with large interaural asymmetries were asked to selectively attend to oddball stimuli in quasi-dichotic presentations of fairy-tale segments, the N400 was abnormal only when they were attending to their non-dominant left ears [102]. The deactivation in the pars triangularis after participating in ARIA could potentially be related to normalization through ascending pathways from the weaker left ears that contribute to enhanced phonological processing of verbal material, but more research is needed to explore that possibility.

Deactivations in the dorsolateral and ventral prefrontal cortices could indicate that these participants required fewer resources from the cortical regions that receive projections from primary auditory cortices and auditory association areas to process dichotic information following ARIA treatment. These areas are primarily involved in detection, discrimination, and working memory and play an important but flexible role in auditory cognition [103]. The region interacts with auditory association cortices to provide executive functions, the processes that go beyond auditory processing per se to guide cognitive tasks [104]. Compared to a control group, children with amblyaudia produced higher levels of neural activity in the right hemisphere during a middle latency response task, especially following input from their dominant right ears [105]. The reduced activity in the right hemisphere in the children in this study after ARIA could potentially reflect a change in the interaural balance of excitation generated by inputs ascending from the two ears. Prior to treatment, excessive dominance in the participants’ right ears during DL tasks could have necessitated increased activation in the right-side dorsolateral prefrontal cortex to detect and discriminate signals arriving from the left ear. Reductions following ARIA could signal that enhanced projections from the left ear may have relaxed demands for detection and discrimination operations in the contralateral right hemisphere. Hypothetically, reduced activity in right hemispheric regions following ARIA could reflect neuroplastic changes in bottom-up sensory brainstem processing of verbal information presented at the listener’s left ear that projects through ascending pathways into the right auditory cortex. Decoding in the auditory cortex is followed by interactions in auditory association regions that send outputs to the dorsolateral and ventrolateral prefrontal cortices on the same side that maintain characteristics of enhanced processing.

The significant improvements seen in dichotic listening and lowering of the signal-to-babble ratio needed for identifying 50% of the single-syllable words in noise from the WIN that occurred for most of the children, including those whose pre-ARIA scores were higher than normal, could be related to the need for fewer working memory and attentional resources after ARIA.

There are several limitations to this study that should be addressed to enhance future research on the outcomes from ARIA for children with dichotic deficits. A larger subject pool, as initially planned for this study, could provide greater power for analyzing the relationships across behavioral performance measures, especially with respect to different speech-in-noise tasks. Comparison of results to a control group is needed to determine if results represent a significant difference prior to enrolling in ARIA and whether changes in brain activity following ARIA are related to the training and not to maturation. Parental reports of difficulties processing auditory input were provided at the time of diagnosis, but were not readministered following ARIA treatment. More evidence is needed to document long-term benefits, including educational outcomes, derived from participation in ARIA training. A caveat to this consideration, however, is that no two children experience amblyaudia the same way, because a binaural integration deficit does not produce identical difficulties in learning, language, or reading across all children. ARIA treatment was developed to enhance auditory access to verbal information and improve neural encoding related to binaural integration. Training-related manipulations of stimulus type and interaural intensities were designed to facilitate bottom-up processing as part of the entire auditory processing system, but because of inherent individual differences and compensatory mechanisms, benefits from ARIA will be highly individualized. ARIA was not developed to directly benefit higher-order processes such as attention, working memory or cognition, but to indirectly aid them through enhanced sensory encoding of binaural verbal information in the auditory brainstem.

Post-training measures were obtained in this study at least 3 months after completing ARIA. Another study investigating immediate changes in neural activation patterns and comparing them to patterns observed after a similar delay period could determine if the effects of ARIA therapy occur immediately following participation or if rest and consolidation contribute significantly to the changes observed in this study. ARIA is not a cure for a DL deficit. The underlying principle behind ARIA is that a large asymmetrical performance is undesirable during DL tasks. Therefore, the primary goal of training is to remove unbalanced levels of excitation and suppression in bilateral pathways ascending through the superior olivary complex to enhance the listener’s access to information presented to both ears during challenging listening tasks. In addition to improved access to DL information, participants in this and previous studies also demonstrated improvements in untrained auditory tasks, including speech in noise, listening comprehension, and word recognition, several months after completing ARIA. Evidence that immediate benefits observed at the final session of training do not regress and that other listening skills also improve over time suggests that enhanced bottom-up access to auditory information from the non-dominant ear may benefit a variety of listening and communication skills. More information is needed to confirm that children with dichotic deficits demonstrate other brainstem-related psychoacoustic weaknesses with nonverbal stimuli and to investigate whether the effects of ARIA are limited to dichotic listening or if other binaural interactions that depend on the auditory brainstem can also benefit from the therapy.

## 5. Conclusions

ARIA training improves interaural asymmetries in children with amblyaudia and may alter both bottom-up and top-down neurophysiologic processes that are apparent several months after completing the intervention. Preliminary evidence suggests that ARIA may facilitate sensory encoding of verbal information presented to the non-dominant ear during dichotic listening tasks and reduce processing demands for top-down cognitive control. Further evidence is needed to substantiate this finding and clarify how auditory training with systematic changes to interaural intensity alters dynamic processes in the brain.

## Figures and Tables

**Figure 1 brainsci-14-00183-f001:**
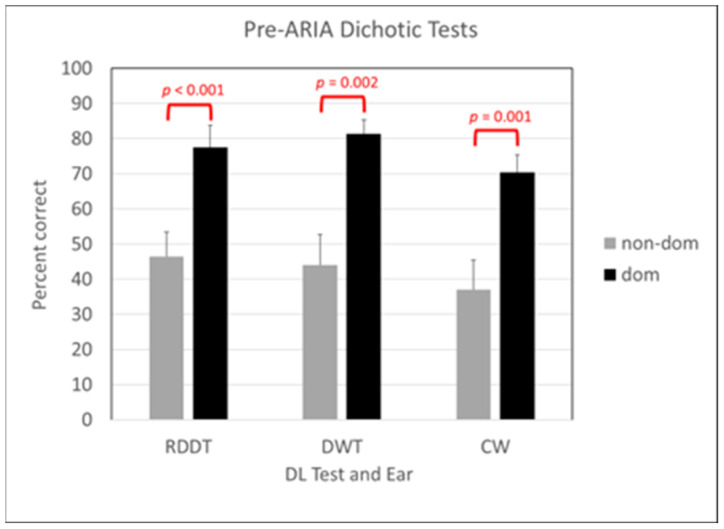
Behavioral test scores prior to ARIA treatment. Scores are in percentage correct for non-dominant ear (gray bars) and dominant ear (black bars) for the randomized dichotic digits test (RDDT), dichotic words test (DWT) and competing words subtest (CW). Significant differences are noted with relevant *p* values.

**Figure 2 brainsci-14-00183-f002:**
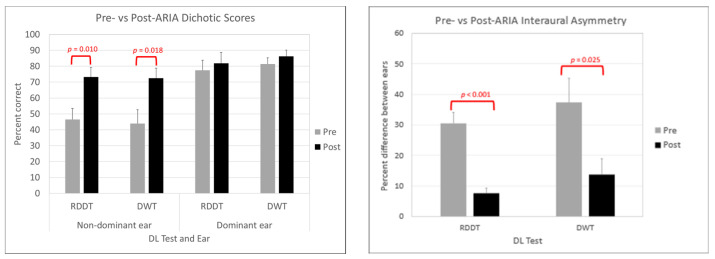
On the left side of the figure, comparison of pre-ARIA (gray bars) and post-ARIA (black bars) percentage correct scores for non-dominant ear and dominant ear for the randomized dichotic digits test (RDDT) and the dichotic words test (DWT). On the right side of the figure, comparison of pre-ARIA (gray bars) and post-ARIA (black bars) percentage differences for interaural asymmetry in scores for the randomized dichotic digits test (RDDT) and the dichotic words test (DWT). All significant differences are noted with relevant *p* values.

**Figure 3 brainsci-14-00183-f003:**
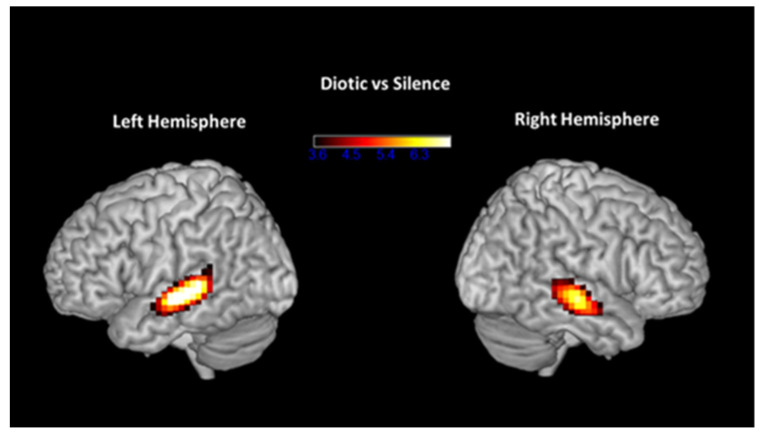
Bilateral temporal regions with greatest levels of activation when comparing diotic condition (listening to the same word in both ears) to silence at baseline, pre-ARIA.

**Figure 4 brainsci-14-00183-f004:**
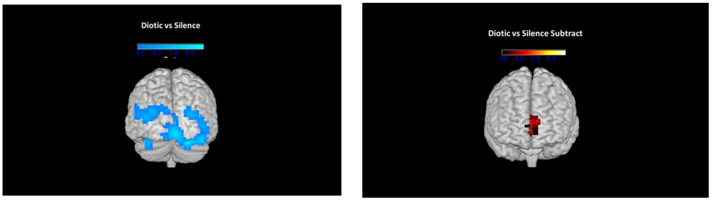
On the **left**, posterior regions with decreased activation when comparing diotic (listening to the same word in both ears) to silence at baseline, pre-ARIA. On the **right**, the same comparison of activation levels during diotic listening compared to silence with activation levels obtained pre-ARIA subtracted from post-ARIA. Positive values indicate an increased level of activation in the anterior regions of the default-mode network when listening binaurally to the same word after ARIA treatment.

**Figure 5 brainsci-14-00183-f005:**
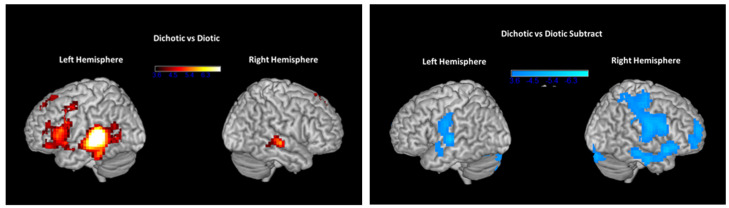
On the **left**, regions with increased activation in left more than right hemisphere during dichotic listening conditions compared to diotic conditions (diotic subtracted from dichotic) before ARIA treatment. These indicate highly asymmetric dichotic-specific activation levels in the left hemisphere prior to treatment with ARIA. On the **right**, the same comparison of activation levels during dichotic listening compared to diotic listening, but with activation levels obtained pre-ARIA subtracted from activation levels post-ARIA. Negative values, therefore, indicate reduced levels of activation for dichotic-specific activation levels following treatment, notably in the right hemisphere.

**Table 1 brainsci-14-00183-t001:** Pre- and post-ARIA behavioral results.

		Pre-ARIA												
		RDDT			DWT									
Code	Age	non	dom	IA	non	dom	IA				WIN	FPT	ID	%ile
2112	7	25	67	42	12	80	68	10	50	40	*9.6*	**7**	AMB	5th
2114	9	61	97	36	88	92	4	73	93	20	8	70	AMB	10th
2120	10	69	97	28	16	76	60	30	73	43	5.6	**20**	AMB	5th
2126	7	27	67	40	28	68	40	30	70	40	*9.6*	**0**	AMB	5th
2128	9	42	86	44	28	92	64	17	67	50	**10.4**	**23**	AMB	5th
2179	8	50	83	33	64	72	8	47	63	16	**12**	**40**	DD	10th
2181	13	86	97	14	72	96	24				4	87	AMB	10th
2183	7	25	47	22	40	64	24	53	77	24	7.2	**0**	DD	5th
2184	9	36	56	20	48	92	44				7.2	**27**	MIX	5th
		**Post-ARIA**												
		RDDT			DWT									
		non	dom	IA	non	dom	IA				WIN	FPT		
2112	7	72	83	11	72	84	12				4.4	**0**		
2114	10	56	61	5	60	60	0				5.2	**27**	DD	5th
2120	10	89	100	11	88	92	4				6.0	**67**		
2126	7	81	97	16	76	100	24				4.4	**40**		
2128	9	58	69	11	64	88	24				5.6	**13**	UND	10th
2179	8	83	100	7	96	100	4				2.4	73		
2181	14	100	100	0	84	88	4				5.8	100		
2183	7	42	44	2	32	80	48				4.4	**20**	MIX	5th
2184	10	78	83	5	80	84	4				6.8	**0**		

RDDT = randomized dichotic digits test; DWT = dichotic words test; non = non-dominant ear; dom = dominant ear; IA = interaural asymmetry; WIN = words-in-noise test; FPT = frequency pattern test; ID = matched pattern identification; %ile = percentile rank for scores; AMB = amblyaudia; DD = dichotic dysaudia, MIX = mixed, UND = unidentified. A red background is for scores representing severe deficits, yellow background is for scores representing moderate deficits, and green background is for scores representing minimal deficits.

## Data Availability

The data presented in this study are available on request from the corresponding authors (privacy restrictions).

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
