# Peer review of "Behavioral and Cortical Activation Changes in Children Following Auditory Training for Dichotic Deficits"

_brainsci, 2024, doi:10.3390/brainsci14020183_

Round 1

Reviewer 1 Report

Comments and Suggestions for Authors

The authors investigated the effect of dichotic listening training on behavioural measures of ear advantage for dichotic speech understanding in children diagnosed with more significant than typical right-ear advantage.

Nine children participated in four one-hour sessions of dichotic listening training in which the intensity for the dominant right ear was adapted according to the task performance.

Behavioural tests of dichotic listening, words-in-noise understanding and spectral pattern recognition were performed before and after completing the training. Also, pre- and post-training fMRI recordings with diotic and dichotic auditory stimuli were performed before and after the intervention.

The behavioural measures demonstrated that participants improved word identification for the non-dominant left ear in dichotically competing listening, thus reducing their otherwise atypically large ear asymmetry. Also, word-in-noise identification was improved significantly in the post-training test. A test of spectral pattern recognition showed pre-training deficits but no consistent post-training changes.

The fMRI analysis showed bilateral auditory cortex activation in diotic listening and left hemispheric dominance with dichotic speech stimuli, suggesting typical left lateralized speech processing in the group of children.

The relatively small sample size of nine children is acceptable given the difficult-to-study population and the enormous logistical efforts required for a training study accompanied by intense training.

Also, the study design as a single-group pre-post comparison is acceptable; however, requires intense discussion of possible confounding factors.

The pre-post training differences in dichotic listening abilities were surprisingly large; even substantial improvement in word-in-noise identification was reported for the intense but relatively short training. These results are encouraging and justify publication.

As a reader, I would prefer a shorter, more focused introduction and a less speculative, shorter discussion; however, these are matters of taste. The value of the manuscript could significantly benefit from a thorough revision of the methods and results section, providing more precise information.

The introduction section is long; however, it provides little insight into how the authors think about (1) the origin of atypical ear advantage for speech understanding, (2) asymmetry as part of the overarching complex of auditory processing deficits, and (3) how the processing asymmetry contributes to communication deficits. It would be helpful to explain to the reader at which degree of asymmetry the typical right ear advantage becomes a deficit. The authors do not develop a clear hypothesis about their training intervention. The link to constraint-induced therapy seems far-fetched because it does not capture the aspect of understanding speech from competing sources. The introduction section should be streamlined and better focus on the training approach.

The methods and results section requires clarifications at several points:

Line 164: ‘Six participants were lost …’ -- Did these children participate in the behavioural tests and the training? If that is the case, the valuable data from these children should be included. Please clarify.

Line 169: Table 1 and the report of the pre-training tests should be moved to the results section. Currently, there is some redundancy between the description of the pre-training test in the methods and again in the results section. The methods section should only report the behavioural measures which were used as inclusion criteria.

Lines 163-182: What was the purpose of the parent questionnaire? Was this used for inclusion/exclusion or evaluation of training effects?

Line 184: In addition to hearing thresholds <25 dBHL, please report the range of between-ears threshold differences.

Line 187: ‘interaural asymmetry was measured as the difference in performance between the two ears’ …. A more common measure would normalize the difference to (e.g.,) the score for the dominant ear or the mean of left and right scores.

Line 188: ‘Ears were identified as “dominant ear” and “non-dominant ear” based on relative performance.’ … what was the criterion for an asymmetry?

Line 191: ‘If abnormal performance patterns did not match across those two tests …’ What was the criterion for matching scores? It is difficult to see consistency between the RDDT and DWT scores in Table 1.

Lines 194-205: The description of how raw scores were transformed into z-scores sounds reasonable. However, it is difficult to apply this description to Table 1. Would it be possible to add the z-scores to Table 1. Also, scores from different tests were used for the diagnosis of an abnormal asymmetry. Would it be possible to indicate in Table 1 (e.g., using bold font) which test was used for the diagnosis?

Lines 206-209: The word-in-noise test seems important – you want to know how central asymmetry relates to word (and sentence) understanding in noise. From Table 1, it appears that the RDDT scores are correlated with the WIN but not the DWT scores.

Line 208: ‘Individual scores were compared to available normative data for both tests.’ …. Where is the outcome of this comparison reported?

Lone 215: ‘The HUSH (hemodynamics unrelated to scanner hardware) technique was used for presentation’ … please explain.

Line 218: ‘fMRI stimuli’ --- ‘auditory stimuli’

Line 225: ‘four weekly one-hour sessions of ARIA’ … That is one session per week for four weeks, a total number of four training sessions for each participant. Is that correct?

Line 230: ‘±90° relative to the participant’s 230 head’ … this means fully lateralized speakers at the right and left ears?

Lines 233-236: It will take some time detecting ear advantages on the order of 10%. Thus, the intensity for the dominant right ear was adjusted for the 20-min blocks of training? Or more frequently?

Line 289: ‘All participants produced a right-ear advantage across DL measures used for diagnosis.’ is produced the right word? We hear for the first time (or did I miss this?), that always the right ear was dominant. Thus, the ear asymmetry was always an enhancement of the right ear advantage found in people with left hemispheric speech processing. This fact could be emphasized. Also the ear of dominance could be indicated in Table 1 (not changed in Table 2).

Line 295: The ear asymmetry is by design, thus, finding a significant difference is trivial. The t- and p- values could be omitted and for clarity, only the effect size reported.

Line 297: ‘Three participants produced abnormally high signal-to-babble ratios for identifying 50% of the words presented in the WIN and two others produced borderline scores.’ --- This description is not consistent with the values reported in Table 1. The WIN scores were almost uniformly distributed between 4 and 12 dB

Line 299: ‘were able to normally identify tonal patterns’ better: ‘were able to identify more than 50% of tonal patterns’

Line 311: ‘intended’ --- expected?

Line 320: ‘Five of the participants produced DL scores at normal levels in the post-ARIA assessment.’ --- please indicate these individuals in Table 2.

Lines 321-330: Again, identify the discussed individuals in Table 2.

Lines 361-363: It is not clear what has been shown in Figure 6. The figure title reads ‘Dichotic vs. Diotic’, which is reasonably the difference in activation between dichotic and diotic conditions. However, the figure caption mentioned a pre-post-training comparison (?) But this is not shown in the figure (?).

Author Response

We thank this reviewer for these helpful comments.  We edited the introduction to make it shorter and more focused. We also modified the discussion section to be as specific as possible about the known primary functions of brain regions that showed changes in activation patterns following our intervention. Our purpose was not meant to state that we could confirm that the changes were definitive but rather suggestive that in these children, brain activation patterns did change in those regions.  But, more is definitely needed to confirm this initial finding which we address in the limitations to the study.  

Modifications to the introduction were made to help clarify mechanisms of dichotic listening, to differentiate between several models and to clarify the brainstem, bottom-up model favored in our research endeavors.  We also expanded information about how constraint-induced therapies work for persons with motor disorders and how our approach utilizes similar methods to achieve reductions in functional asymmetry in the auditory system. 

We elected not to include the data from the 6 children who were not able to participate in the final fMRI scan because we felt that the behavioral outcomes should be limited only to those who were able to complete the entire study.  We have confirmed that their initial behavioral results were consistent with dichotic deficits and that they, too, demonstrated behavioral benefits after treatment.  

We combined the results from Table 2 into those in Table 1 to include only one table.

We explained the purpose of the parent questionnaire.

We noted that no participants had interaural threshold differences > 5 dB HL.

We expanded our explanation of interaural asymmetry and application of cut-off scores to determine when a participant's results are deficient.

We noted the age dependency of asymmetry values for dichotic tests.

We did not evaluate correlation with WIN scores - but only noted the improvement seen across most of the participants, including those with borderline or abnormal scores whose scores were normal after participating..

We clarified the structure of the ARIA protocol.

We put in subject numbers for participants referred to in the discussion section. 

We rewrote the figure legends for the scanning images to help clarify what is being measured.  

Reviewer 2 Report

Comments and Suggestions for Authors

The research deals with behavioral test performance and different fMRI cortical  activation after Auditory Rehabilitation for Interaural asymmetry training in a group of 9 children with dichotic listening deficit . The topic is relevant and to the authors must be given the merit of conducting the research study in a rigorous and impartial way. 

The introduction provides the concepts necessary to contextualize the study and have an understanding of the research topic. However, the difference between dichotic and diotic listening is clarified in the results at lines 364, 365 and 341,342, respectively. It could have been useful clarify this difference in the introduction.

Materials and methods, and results are described in detail, while the discussion and conclusions do not include unsupported suggestions. Limitations and future areas of research are clearly spelled out.

Despite the use of complex and structured English, with compound-complex sentence structure, and the presence of some misprints that can however be corrected during editing, the manuscript is worthy of publication in the present form.

It will obliviously be the authors' choice whether to make the following adjustments:

- the abstract in lines 10-13 appears unclear regarding study design (monocentric observational study?), participant numbers, and selection criteria. The latter two not required to be reported. It may be helpful to replace "one group pretest-posttest design, parent report of listening difficulties," with "monocentric observational study"

- "see Table 1 in [13]" in line 107, "ollowing" in line 149, "see Methods below" in line 159 and "see below" in line 191 should be removed or corrected

- Remove misprints in lines 169, 242, 300, 331, 353, 375

- Table 1 shows results that will be discussed in the text in the next paragraph and reported in addition in Table 2. It might be useful to create one larger table to be reported on page 5 or to separate the epidemiological features and results in two different tables.

- Figures 2 and 3 could be merged into one figure 2 (A and B) with a single caption. Similarly, Figures 4, 5 and 6 and Figures 7 and 8 could be merged in two different figures.

Author Response

We thank this reviewer for these comments.

We clarified the difference between diotic and dichotic in the introduction and throughout the manuscript, especially in the results section.

We are not familiar with the term monocentric for a research design.

We combined the results into one Table as suggested.

We also merged the figures as suggested.  

Reviewer 3 Report

Comments and Suggestions for Authors

The paper can be interesting, to its 'audience', and the findings can be considered valuable. 

The Introduction is ok, but should stress more on 1) the significance of the article in its field of studies and 2) the specific research goals and how the Authors plan to achieve them. 

The paper needs a proper Literature Review, possibly in a section between the Introduction and the part on the results. As it is now, the literature review itself is incomplete and 'scattered' here and there, across the article, and that makes this work not completely up to academic standards. 

The methodology is ok, and this Reviewer can follow it, but 1) should be expanded and 'stream-lined', also clearly separating it from the 'materials', and 2) should be double-checked, for final validation, by a Reviewer with a more direct expertise than me in the topic analyzed (the same Reviewer should also double-check and validate the figures). 

I am not completely sure that the number of partecipants - the 'sample' - is consistent enough, quantitatively, to provide the Authors and the readers with indicative results, but that should be 'decided' by a Reviewer with a more specific expertise than me in the topic of the paper and, of course, by the Editors. 

The Results section is ok, 'linear' enough and relatively 'user-friendly'. 

The Discussion is quite comprehensive. Perhaps the Authors could add more 'analytical moments', especially focused on their own interpretations of the results (I mean, the more we have observations and remarks by them, the better and richer is the article). 

The Conclusions are incredibly short - too short. They should 'wrap up' the paper, like a closing 'mini-discussion', and, like in a 'mirror' with the Introduction, should stress again on the importance of the findings of the paper in its panorama of research and on how the Authors achieved their goals (i.e., their strategies and technical implementations). 

All in all, the paper can be considered interesting and noteworthy, but it requires a thorough revision before being qualified to be considered for publication. 

Thank you very much. 

Comments on the Quality of English Language

The English language is fine and professional. 

  However, the written style is a little 'robotic' and could be improved, making it, so to speak, 'smoother', to allow it to flow better. 

Author Response

We thank the reviewer for these comments.

We modified the introduction to focus more on specific research goals. 

We also modified the discussion to focus primarily on known functions in brain regions wherein we reported changes following treatment.  The purpose is to note that these are relevant brain regions for the tasks that we were assessing behaviorally and to suggest that our results may provide some preliminary evidence that the neural resources required to perform the tasks may have changed following treatment. 

We kept the conclusions section short because we wrote it to be a brief summary of what we covered in the discussion and focus on what should come next. 

Round 2

Reviewer 1 Report

Comments and Suggestions for Authors

I reviewed a previous version of the manuscript and suggested several clarifications. The authors responded in detail to the questions and revised the manuscript.

I still find it disadvantageous that six children were not included in the analysis of the behavioural data because their post-training MRI recording could not be performed for technical reasons. The study does not contain a formal comparison between behavioural and brain imaging findings. Thus, different numbers of participants for both methods should be no problem. However, analyzing 15 instead of 9 participants would have increased the significance of the behavioural data.

Also, as a reader, I would have been interested in a correlation between the results of dichotic listening tests and speech in noise understanding.

But then, I can accept the author’s decisions about their focus on the study.

Overall, I am satisfied with the revision of the manuscript. There are only a few minor details reqiring further attention:

Abstract:

line 10: ‘quasi-experimental’ – could be removed

line 12: change ‘several’ to ‘3 to 4.5’

Introduction:

line 37: ‘Kimura’s structural theory of DL [11] attributed the DL asymmetry to poor neural transmission through the corpus callosum following verbal input to the ear that is ipsilateral to the listener’s language-dominant hemisphere.’ could be changed into a positive form to avoid the impression of a general deficit of ‘poor transmission’ in the ipsilateral pathway. e.g.: ‘Kimura’s structural theory of DL [11] attributed the preference for verbal input to the ear that is contralateral to the language-dominant hemisphere to stronger transcallosal transmission in the contra-lateral pathway.’ The deficits in Kimura’s patients were temporal lobe lesions but not abnormal callosal transmission. The original reference to the asymmetry in crosscallosal connectivity would be Rosenzweig 1951.

line 48: ‘An attentional model …’

line 95: not clear what ‘by aligning 95 lists of dichotic material’ means

line 100: The sentence beginning with ‘CIT utilizes forced use to encourage …’ could be polished.

Methods:

line 145: remove ‘(see Methods below’)

line 223: this reference to the table could be removed

line 271: …. measures. Six …’ also consistent usage of numbering words

line 280: Table 1 (also elsewhere)

line 280: please quantify ‘borderline’, I could not find italic numbers for the WIN scores in the table

Line 318: The table caption is missing

Comments on the Quality of English Language

The manuscript could benefit from minor polishing.

Author Response

Thank you again for your comments. 

I do have the pre- and post-ARIA data from the other 6 children and when I run them with the 9 from the study as reported, the significance of the difference in non-dominant ear scores and interaural asymmetry increases with a drop in the value of r2 likely because there is more variability with the additional children included.  Here are the results from evaluating all 15: For non-dominant ear on RDDT and DWT, F (1,29) = 9.325 and 9.395 respectively, p = .005 in both cases, and r= .25 and .22 respectively.  For interaural asymmetry, F (1, 29) = 20.3, p < .001, and r= .40 for RDDT and F(1, 29) = 7.68, p = .010, r=.18 for DWT.  I did not add them to the manuscript, though, because I prefer to only include the children from whom the images were obtained.  

You also asked about the correlations between dichotics and words-in-noise so I ran those and did include that information in the manuscript.  

I also went through your individual recommendations and made the changes you suggested with these exceptions:

line 145 "(see Methods below)' was not in the current version

line 271 I did not see the ....measures. Six....' 

line 280 is a Figure 

Reviewer 3 Report

Comments and Suggestions for Authors

Thank you for the revision. 

  Not all my comments have been addressed. 

  Especially, you ignored completely my suggestion inherent in the Literature Review. 

  However, I still think it needs significant improvements, at least at the level of format. 

Author Response

Thank you again for your comments.  

I am sad that you thought your input had been ignored.  I revised the Introduction substantially to make it a comprehensive review of the literature underlying the models of dichotic listening proposed by early proponents of the tests, current thinking about the neurophysiology involved in dichotic listening processes, and evidence of deficits in dichotic listening observed in children with a variety of learning difficulties.  For the therapy used in the study, I detailed the basic principles behind the training approach to help the reader understand that our imaging study's primary aim was to evaluate whether the training did alter the neurophysiology in the brains of the children who had enrolled in it. My intent in the review of the literature was to set the stage for the primary aims of the study and to limit its scope to the fundamental aspects of prior literature that were most directly relevant.  Please advise if there is another approach that I should have taken.  

You noted a need to improve cited references - but did not indicate whether you think some should be removed or others should be added and for what purpose.